# Properties of Lightweight Insulating Boards Produced from Triticale Straw Particles

**DOI:** 10.3390/ma16155272

**Published:** 2023-07-27

**Authors:** Mariusz Lesiecki, Jakub Kawalerczyk, Adam Derkowski, Marek Wieruszewski, Dorota Dziurka, Radosław Mirski

**Affiliations:** 1Center for Innovation and Transfer of Technology, Poznań University of Life Sciences, 60-627 Poznań, Poland; mariusz.lesiecki@up.poznan.pl; 2Department of Mechanical Wood Technology, Faculty of Forestry and Wood Technology, Poznań University of Life Sciences, 60-627 Poznań, Poland; jakub.kawalerczyk@up.poznan.pl (J.K.); marek.wieruszewski@up.poznan.pl (M.W.); dorota.dziurka@up.poznan.pl (D.D.)

**Keywords:** insulating materials, straw particles, lightweight boards, thermal insulation properties, acoustic insulation properties

## Abstract

Insulating materials made from straw are becoming increasingly popular in the construction industry. Straw can be used in the construction of buildings as uncompressed straw chips or in the form of compressed panels. This study aimed to determine the possibility of manufacturing boards from straw particles with densities in the range of 150–400 kg/m^3^, allowing favorable mechanical properties while simultaneously providing high thermal and acoustic insulation properties. The study also analyzed the influence of the degree of carpentry density on the quality of the manufactured boards. The study shows that insulation boards can be produced from straw particles with satisfactory properties already at densities in the range of 200–150 kg/m^3^. Boards with this density have a compressive strength of 150 kPa, thermal resistance of 0.033–0.046 W/(m·K), and a sound absorption coefficient above 0.31.

## 1. Introduction

Cereal straws are a secondary material or even a production waste in producing seed (grain) for food or animal feed. The intensification of agricultural production currently does not allow the full use of straw for animal husbandry purposes, resulting in a worldwide overproduction of straw each year, which is used less rationally than its chemical composition and structure would allow. The chemical composition and physical structure of straws, especially of rye, wheat, or triticale, are similar to wood. Currently, technologies are increasingly being developed to use straw as a substitute for wood in the production of panels for use in construction. The literature describes many methods of using straw as a substitute for wood or even as the sole raw material in producing panels intended for furniture making [1,2,3,4,5,6]. Various methods of processing straws and other lignocellulosic materials are also known, leading to the use of these materials in the construction industry both as insulating and structural elements [7,8,9,10,11,12,13,14,15].

From description of a patent provided by Tsutomu and Masanao [16], it appears that straw panels can be used for soundproofing a multi-layer floor. Another patent description, on the other hand, shows that it is possible to produce a composite straw panel with high mechanical properties by combining restructured bamboos and the straws in three-step production technology [17]. Furthermore, rice straw or, more precisely, straw particles of 30–50 mm in length can be used to produce thin boards with reduced formaldehyde content by applying isocyanate glue in the amount of 4–6% [18]. This type of board can be a good substitute for thin MDF or HDF boards. An attractive solution is sandwich boards, where a middle layer of straw is sandwiched between wood-based panel facings [19]. It is clear from the description that the straw should be straight; i.e., it should not have been baled or baled beforehand, as otherwise, it is impossible to control the optimum fiber length (11 mm) and its orientation in the panel structure. The latter operation is, moreover, challenging and complicated. Studies performed by Wei et al. [20] confirmed that the length of the straw has a significant effect on the properties of resultant board. The authors stated that reduction in the dimensions of applied rice straw contributes to the improvement in mechanical properties of the panels; however, it also negatively affects their insulation characteristics. Another solution described by Zhu et al. [21] faces a similar problem. In this case, long strands characterized by the length of 30–80 mm are required to obtain good mechanical properties of the boards. Pressing results in high density boards. Such boards do not have good acoustic properties but have increased strength. Yang and his team obtained boards with good acoustic properties [22]. The boards they produced had sound absorption coefficients in the range of 0.4–0.6 but were more structural than insulating, with densities in the range of 400–800 kg/m^3^ and static flexural strengths of approximately 10 to 20 N/mm^2^.

The solutions presented above are most often characterized by the manufacturing of boards with an average density of more than 500 kg/m^3^. However, the thermal insulation of the boards increases inversely proportional to their density, so materials with a density below this value, usually even below 300 kg/m^3^, are used as insulation materials. Attempts to produce panels with low densities below 180 kg/m^3^ and low thermal conductivity coefficients of 0.048–0.036 W/(m·K) were made by Harvey in 1979 [23,24]. This should be considered quite an achievement, as the study used ground straw with an average length of 0.25”, or about 6.4 mm. In contrast, Hussein’s work has reported studies over a wide range of densities, from 300 to 700 kg/m^3^ [25]. The physical–mechanical properties of boards with a density of 300 kg/m^3^ should be considered good, providing potential for further research. Good properties of boards with similar densities can be obtained by high-frequency current pressing [20] or by sealing the straw with cement [26].

The fabrication of these types of lightweight panels poses many difficulties. These are due to the way the straw is sourced and the technology used to manufacture the panels. Straws are most often obtained by baling or pressing [27]. In the former case, they are baled into a cylindrical shape, and in the latter case, they are pressed into a cube. This process largely crushes or breaks the stalks, causing them to be destroyed, and it does not produce long chopped straw (chips) with a regular structure, as would be the case if the straw were harvested by hand (with pickling machines). Therefore, taking into account the following hypotheses that the applied dimensional fraction of straws has a major influence on the insulation characteristic and that the triticale straw can be used as a valuable material for insulation boards manufacturing, this study aimed to evaluate the possibility of producing triticale-based lightweight boards intended for thermal and acoustic insulation from straw particles/chips produced from straw previously baled (a method that makes it difficult to obtain sufficiently long fibers, as required in insulation boards).

## 2. Materials and Methods

Triticale *(X Triticosecale Wittmack)* straw was used in this study, as not only from the name but also from macroscopic studies, it appears that this straw has characteristics similar to both rye (length of elements) and wheat (thickness of elements) [28]. The straw was ground with a disc chopper (Research & Development Centre for Wood-Based Panels, Czarna Woda, Poland) to particles of linear dimensions, allowing bulk densities of 25 kg/m^3^ (variant A) and 50 kg/m^3^ (variant B). The straw particles prepared in this way were used to produce board formats under semi-technical conditions with average densities of 150 kg/m^3^, 200 kg/m^3^, 250 kg/m^3^, 300 kg/m^3^, 350 kg/m^3^, and 400 kg/m^3^. Boards made from type A particles were produced using a gluing ratio of 5% (adhesive dry weight to dry straw weight), while type B particles were glued at 10%. The lignocellulosic material was glued with pMDI glue (Bayer AG, Leverkusen, Germany). The gluing operation was performed with pneumatic system LFG-5 (Devilbiss, Warsaw, Poland). The moisture content of the straw before pressing was 8.53%. The pressing parameters previously selected based on their history in working as insulation materials were adopted: temperature of the heating plates of 200 °C, pressing time of 20 s/mm, and pressing pressure of 0.8–1.2 MPa. During the pressing test, the temperature in the center of the mat and on the top surface of the mat was measured. The measurements were made with a multimeter using a thermocouple K. After completion of the two-week conditioning process (temperature 20 ± 1 °C and relative humidity of 65 ± 5%), the produced boards were evaluated in terms of following properties:-The assessments of density and short-term water absorption were carried out on samples with dimensions of 100 × 100 mm and thickness of the boards produced, i.e., 25 mm. The density assessment was carried out based on EN 323 [29], while the short-term water absorption was carried out based on EN ISO 29767 [30].-The compressive strength and compressive modulus were assessed on specimens measuring 50 × 50 × 25 mm. The compressive stress value was assessed at ten percent relative deformation (EN 826 [31]).-The evaluation of tensile strength perpendicular to the planes of the board was carried out on samples measuring 50 × 50 × 25 mm (EN 319 [32]).-Evaluation of the thermal conductivity coefficient was carried out on samples measuring 220 × 252 mm with a manufactured board thickness of 25 mm. The tests were conducted at an assumed temperature of the warm medium of 60 °C. Measurements were made using J-type thermocouples (Thermo Aparatura, Wrocław, Poland). The design and principle of operation are presented in the recent work of Mirski et al. [33].-The evaluation of the sound absorption coefficient was carried out on circular samples of 100 mm and 30 mm in diameter. The sound absorption coefficient was determined in an impedance tube by EN ISO 10534-2 [34]. The tests were carried out by BOSMAL (Bielsko-Biała, Poland).-Evaluation of the density profile on the cross-sectional area was carried out on samples measuring 50 × 50 mm using a DAX 6000 laboratory profilometer (GreCon, Hannover, Germany).

The mechanical properties were assessed using a Tinius Olsen 10 KN testing machine (Tinus Olsen, Redhill, UK). The tests were mostly carried out on 4 specimens of 100 × 100 mm and 6 specimens of 50 × 50 except of the density profile measurement. In this case, the measurement was made only on 3 samples, as were the acoustic and thermal insulation measurements. The dedicated software Statistica 13.0 (Version 13.0, StatSoft Inc., Tulsa, OK, USA) was used for the statistical evaluation of the test results obtained. The data were subjected to ANOVA tests which are an analysis of variance used to analyze the differences between variants. Moreover, the post hoc Tukey HSD test was used to obtain the homogeneous groups.

## 3. Results and Discussion

Long-standing research in the department shows that by using a grinding mill to obtain wood chips for the middle layer, a bulk density of the particle cake of about 60 kg/m^3^ is obtained from the straw stalks passing through the mill twice [35]. Therefore, for variant A, the straw was passed through the mill only once, and the fraction remaining on the 10 × 10 mm mesh sieve and passing through the 1 × 1 mm mesh sieve was sorted from the chopped straw obtained. However, the straw was milled twice for variant B, and the fraction passing through a 1 × 1 mm mesh sieve was discarded. The result for variant A was a higher proportion of larger fractions (those remaining at least on the 4 mm mesh sieve) at the expense of the finer fractions (Figure 1). The situation in the material of variant B is the opposite.

However, both variants are dominated by fine fractions, which contain more than 63% chopped material in variant A and more than 85.5% in variant B. For both variants, however, the bulk densities obtained were similar to those assumed, i.e., for variant A: 24.3 ± 2.1 kg/m^3^, while for variant B: 48.1 ± 2.7 kg/m^3^.

The length of the strips is relatively large (Figure 2). The straw strips remaining on the 6.3 mm mesh sieve are over 40 mm long; the fractions remaining on the 5.0, 4.0, and 2.5 mm sieves are about 25–27 mm; and those remaining on the 1 mm mesh sieve are about 15 mm.

During the pressing tests, the temperature change was determined in the center of the mat and on the surface. The results obtained for the three measurements were averaged, resulting in sharper changes in the graphs than the temperature change process occurred (Figure 3). However, some differences in the overheating process of the mats for the assumed densities of 150 kg/m^3^ versus 300 kg/m^3^ can be seen quite clearly. The mat overheated much more slowly after compression to 150 kg/m^3^ than the mat with a two times higher density. Both variants reach a similar temperature range at the center of the board, approximately 160–170 s after the start of compression. It is followed by a faster temperature rise above 100 °C for the lighter board. The intensity of the overheating of the mat is highly dependent on the moisture content of the material being pressed [36]. It should be noted that the susceptibility of the mat to vapor removal also influences its overheating and the achievement of a temperature in the middle zone that allows the glue to cure fully.

When the boards were manufactured, it became apparent that those made from variant B, with an assumed density of 150 kg/m^3^, were of very poor quality, so an evaluation of their properties was omitted. The manufacture of boards with a density of 400 kg/m^3^ from type A particles was also omitted. As can be seen from the data in Table 1, the produced boards, even with a low density, are characterized by high mechanical properties. The most important of these properties is, of course, the compressive strength. This is because it determines the material’s resistance to external loads that cause the structure of the material to compress, which is unfavorable for materials used in wall or floor insulation. The compressive strength range of 150–200 kPa corresponds to the quality of polystyrene foam used for insulating floors or car parks. Compared to an insulation board made of fiber mass, the resistance of straw boards with a similar density is almost three times higher. As the density increases, the compressive strength of the boards increases exponentially (fc = 21.27exp(0.0112ρ) R^2^ = 0.9861).

Furthermore, as can be seen from the data in Figure 4, there are no significant differences in the compressive strength of the boards in the density range of 200–350 kg/m^3^. However, boards made from a mass with twice the compression ratio have a slightly higher compressive strength. The modulus of elasticity, especially for boards with a density of less than 250 kg/m^3^, is not high. This means that the material’s permanent deformation will occur due to even low pressure. The tensile strength perpendicular to planes is not a critical parameter for insulation materials. However, the values obtained make it possible to assess the compactness of the straw boards, which show high compressive strength and relatively high tensile strength at a relatively low density. Previous research results have shown that at densities above 650 kg/m^3^ can obtain straw chopped boards with the characteristics of furniture boards [35,37,38].

Good physical properties also characterize the manufactured boards. Thus, short-term water absorption ranges from 1.34 kg/m^2^ for the lightest boards to 2.15 kg/m^2^ for the boards with the highest assessed density (Table 2). Compared to the recommendations, these values are almost two times higher, as it is assumed that short-term water absorption tested by the partial immersion method should be below 1 kg/m^2^. This assumption is met by boards made from wood particles (Table 2) and by insulating materials made from lignocellulose pulp extracted from straw [39]. However, it should be noted that the industrial boards are protected with additional hydrophobic agents, either on the surface or in the mass, making them more resistant to moisture. In contrast, no additional water-resistant agents were used in this study.

The thermal conductivity coefficient of the manufactured boards is relatively low and on the level of commonly used insulation materials made from lignocellulosic raw materials. Interestingly, the boards made from Type B particles have lower thermal conductivity coefficients than those made from Type A ones. One could try to explain this fact by variations in the quality of the chips and thus in the way they are arranged in the board structure. However, from the analysis of the density distribution over the board cross-section, no clear differences in the shape of the profiles of the two board types can be seen Figure 5). Boards with a density close to 150 kg/m^3^ show an almost flat cross-section, with no clear compaction of the near-surface layers. Some changes in this respect are only observed when the density of the boards approaches 250 kg/m^3^. It should be remembered that the boards that are manufactured from chips/sawdust with a low moisture content favor compaction of the near-surface layers. However, the classic arrangement is only observed at a density of 400 kg/m^3^. Thus, another important difference in preparing these boards is the degree of sealing. Its influence is observed in the analysis of absorption and compressive strength. The chips that are more bonded show significantly better properties. The more intense, the higher the density analyzed. However, the boards with a higher degree of bonding show better insulating properties. Although it is not visible in the runs of the density profiles, the degree of fragmentation of the chips and their arrangement in the board determines, in this case, the thermal characteristics of the boards produced. Thus, a more fragmented material allows for better mechanical properties of the panels, while less fragmented material allows for better insulating properties (thermal and acoustic). This is consistent with the observations of other researchers [40].

The study also shows that only boards with a density of 150 kg/m^3^ have a very high sound absorption coefficient (Table 2). In the case of boards with higher densities, the values obtained for this coefficient, although several times higher than for furniture or construction boards, are significantly lower than for acoustic insulation materials [41].

In contrast, there were no significant differences in the sound absorption coefficient values for panels with densities above 250 kg/m^3^ made from different straw chopped straw variants. Figure 6 shows the appearance of the samples for the evaluation of the acoustic characteristics, and Figure 7 shows the detailed results of the measurements.

As can be seen from the data in Figure 5, straw boards with a lower bulk density are more effective in absorbing sounds in the 1000–1600 Hz and 4000 Hz ranges. In general, at lower frequencies, boards made of finer chopped straw absorb sound better. In contrast, what is a very important difference that distinguishes straw panels from insulating panels is that they insulate sound better up to a frequency of 1250 Hz. In comparison, panels made of glass wool do so for frequencies above 1250 Hz.

## 4. Conclusions

Research has shown that fine straw boards can be a good alternative to other thermal and acoustic insulation materials. In addition, straw boards have a relatively high compressive strength, allowing them to be used more widely and as a subfloor material. The test results obtained can be described as follows:
-All boards, irrespective of density, are characterized by a very high compressive strength. Compressive strength of 200 kPa is considered very high.-The boards produced are characterized by a relatively low modulus in compression, with better values obtained when more finely chopped straw is used. This is probably due to the lower susceptibility to sticking of long particles than short particles. However, the property in question does not significantly impact the performance properties. The long stripes compensate very well for the lower glue consumption by half for the other properties.-Despite their low density, the manufactured boards have a very high tensile strength perpendicular to the planes of boards. Values higher than 0.04 N/mm^2^ guarantee high resistance to this type of deformation. This type of deformation rarely occurs with the standard use of insulation materials.-Despite the absence of additional hydrophobic agents, the manufactured boards show very low water absorption. This absorptivity increases only slightly as the density of the board increases. It is, however, significantly lower than a mineral wool panel, although it is higher than a reference panel made from lignocellulose pulp. If this cannot be compensated for in the industrial production of boards made from straw particles, additional hydrophobic agents will have to be introduced into the manufacturing technology.-Straw particle boards manufactured in a proposed way show high insulation properties, both in terms of thermal and acoustic insulation.-Larger particles (Variant A) allowed boards with better thermal and acoustic resistance properties to be obtained. On the other hand, a more finely divided straw, with the mass of smaller particles (variant B), allows boards with better mechanical properties to be manufactured. For these reasons, the finer particle boards are suitable as load-bearing insulation boards, e.g., for floors, while the larger-particle boards are suitable as wall insulation boards.


## Figures and Tables

**Figure 1 materials-16-05272-f001:**
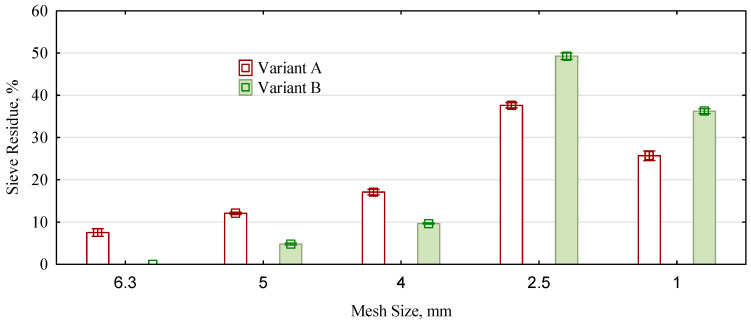
Fractional composition of the investigated materials.

**Figure 2 materials-16-05272-f002:**
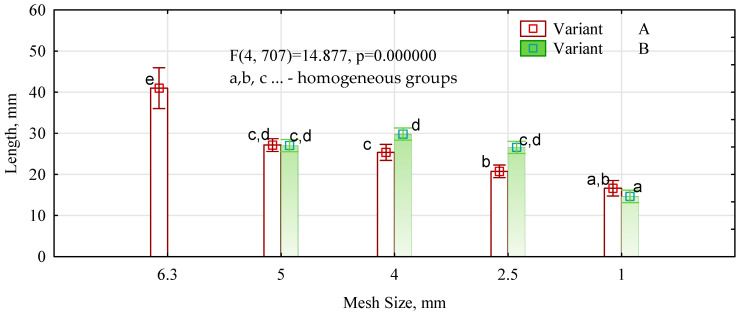
Length distribution of the chopped straw used.

**Figure 3 materials-16-05272-f003:**
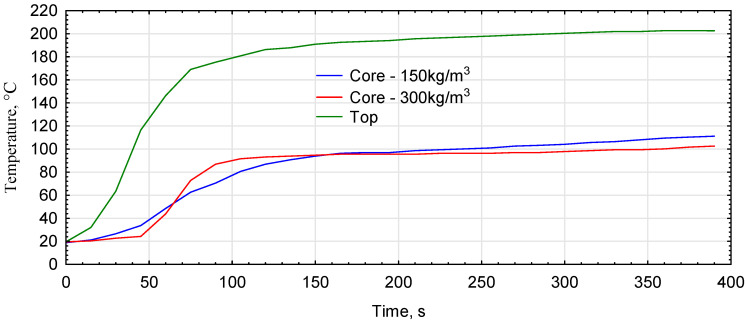
Temperature changes in the structure of the mats during pressing.

**Figure 4 materials-16-05272-f004:**
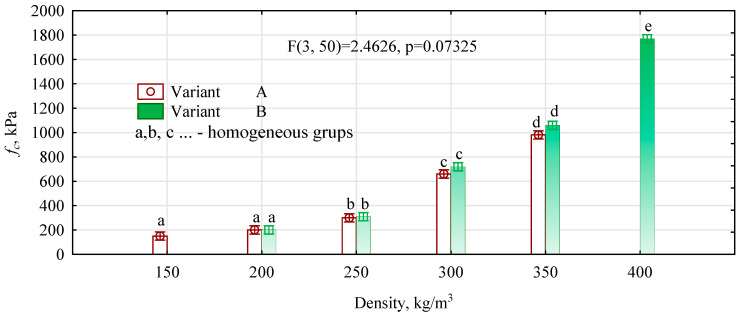
ANOVA analysis of compressive strength—interaction diagram.

**Figure 5 materials-16-05272-f005:**
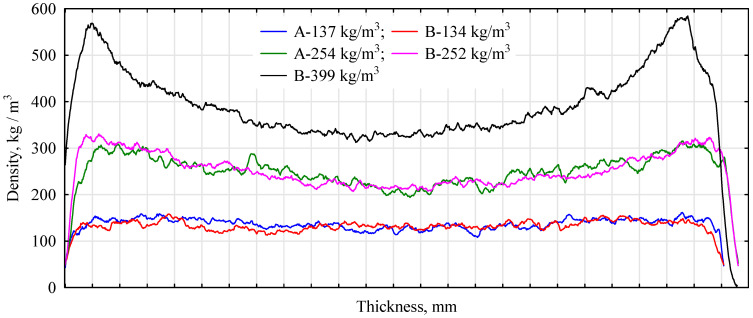
Examples of density profiles of manufactured insulation boards.

**Figure 6 materials-16-05272-f006:**
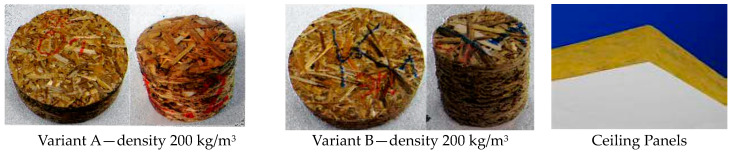
Appearance of the samples intended for the determination of the sound absorption coefficient.

**Figure 7 materials-16-05272-f007:**
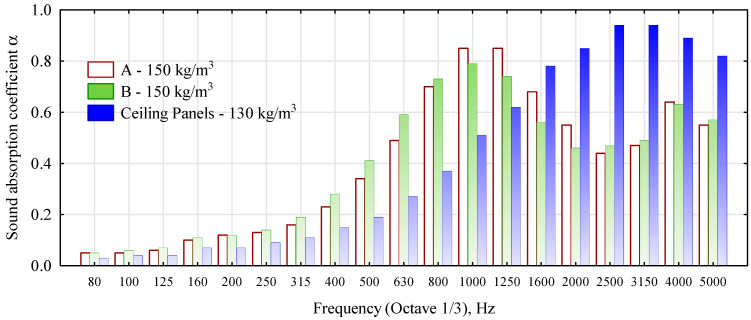
Detailed distribution of sound absorption coefficient for selected material.

**Table 1 materials-16-05272-t001:** Mechanical properties of single-layer panels made from straw particles.

Board Density (kg/m^3^)	Compression Strength (kPa) *		Modulus of Elasticity in Compression (N/mm^2^)		Tensile Strength (N/mm^2^)	
-	A	B	A	B	A	B
150	150 (8.6)	-	0.132 (0.004)	-	0.02 (0.001)	-
200	201 (11.2)	200 (6.0)	0.167 (0.004)	0.214 (0.004)	-	0.04 (0.004)
250	300 (25.6)	310 (26.1)	0.201 (0.005)	0.291 (0.005)	-	0.05 (0.008)
300	660 (38.6)	720 (23.0)	0.512 (0.015)	0.743 (0.010)	-	0.06 (0.007)
350	980 (66.2)	1060 (34.8)	0.950 (0.012)	1.07 (0.011)	-	0.10 (0.011)
400	-	1770 (105.9)	-	1.91 (0.106)	-	0.15 (0.027)
130 **	12.3 (0.02)		-		0.01 (0.001)	
240 ***	100 (2.3)		-		-	

*—fv compressive stress value at ten percent relative strain. **—mineral wool ceiling panel, *** board made of lignocellulosic fiber and cellulose fibers glued with pMDI, manufactured according to PN-EN 13171—reference board.

**Table 2 materials-16-05272-t002:** Physical properties of single-layer panels made from straw particles.

Board Density (kg/m^3^)	Short-Term Water Absorption (kg/m^2^)		Thermal Conductivity Coefficient λ W/(m·K)		Sound Absorption Coefficient α ***	
-	A	B	A	B	A	B
150	1.34 (0.04)	-	0.037 (0.001)	-	0.54	0.52
200	1.66 (0.06)	1.79 (0.23)	0.046 (0.001)	0.033 (0.002)	0.31	0.31
250	1.82 (0.08)	1.82 (0.01)	0.054 (0.001)	0.039 (0.001)	0.18	0.19
300	2.00 (0.07)	1.91 (0.05)	0.063 (0.003)	0.047 (0.001)	-	0.16
350	2.09 (0.07)	1.96 (0.07)	0.072 (0.001)	0.055 (0.002)	-	0.08
400	-	2.15 (0.18)	-	0.068 (0.004)	-	0.06
130 *	4.5		0.031		0.58	
240 **	<1		0.050		-	
180 ****	<1		0.048		-	

*—mineral wool ceiling panel, ** lignocellulose fiber board and pMDI bonded cellulose fiber board manufactured to EN 13171—reference board, ***—average sound coefficient for frequencies: 250, 500, 1000, 1250, 2000, 2500, and 4000 Hz, ****—INTERNAL board as per manufacturer VestaEco.

## Data Availability

The data presented in this study are available on request from the corresponding author.

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
