# Peer review of "Properties of Lightweight Insulating Boards Produced from Triticale Straw Particles"

_materials, 2023, doi:10.3390/ma16155272_

Round 1

Reviewer 1 Report

(1) It is important to add the research progress on the relationship between straw length and the insulation properties of the boards in the introduction, consider the following papers: http://dx.doi.org/10.1016/j.enbuild.2014.11.026, https://doi.org/10.1016/j.susmat.2019.e00102. 

(2) The authors mentioned that "Boards made from type A particles were produced using a gluing ratio of 5% (adhesive dry weight to dry straw weight), while type B particles was glued at 10% in materials and methods. Why the amount of adhesive chosen is different? Whether different amounts of adhesives affect the properties of the material, such as compressive strength.

(3) It is necessary to comprehensively evaluate the boards prepared with variant A and variant B  in the conclusions.

(4) It is suggested that the authors add the bending resistance and thickness swelling of the straw boards, which is also of great significance for the application of the material.

Author Response

Dear Reviewer

thank you for your help in editing this publication.

(1) It is important to add the research progress on the relationship between straw length and the insulation properties of the boards in the introduction, consider the following papers: http://dx.doi.org/10.1016/j.enbuild.2014.11.026, https://doi.org/10.1016/j.susmat.2019.e00102.

The relationship between the insulation properties and straw length is commented based on the paper you mentioned in the introduction section as you suggested.

(2) The authors mentioned that "Boards made from type A particles were produced using a gluing ratio of 5% (adhesive dry weight to dry straw weight), while type B particles was glued at 10%” in materials and methods. Why the amount of adhesive chosen is different? Whether different amounts of adhesives affect the properties of the material, such as compressive strength.

As the bulk density increases, the specific surface area of the particles increases. This increase is not always in a 1:1 relationship but can be assumed with some approximation. Therefore, straw particles with a higher bulk density (twice) were glued with two times the glue content. This is an essential technological procedure which, of course, is always finalised on the processing line. I agree that a higher amount of glue, in this case not to weight, but to surface area, will affect the mechanical characteristics.  

(3) It is necessary to comprehensively evaluate the boards prepared with variant A and variant B  in the conclusions.

Thank you for your valuable suggestion. It is now included in the text.

(4) It is suggested that the authors add the bending resistance and thickness swelling of the straw boards, which is also of great significance for the application of the material.

Thank you for your valuable suggestion. Unfortunately, we no longer have these boards to conduct additional studies. The scope of work was developed in consultation with representatives of the industry involved in the production of insulation materials. Our common view is that these properties are less important for these types of materials, however we agree that they are also worth investigating in future research. Compressive strength is the main mechanical characteristic for insulation materials and their assumed use does not involve exposure to water.

Reviewer 2 Report

The topic of the research work and manuscript is really interesting and provides new and useful information. However there are some issues to be addressed towards its quality improvement before publication. In the abstract, you refer "thermal insulation of 0.033 - 0.046 W/(m-K)" and I wonder if the term thermal resistance would be more appropriate to be used. In lines 35-38, as well as in the points that you refer to a patent or a study, please provide the most crucial and relevant to your topic findings (or for example in line 40, you refer reduced formaldehyde content without providing the reader how this was achieved). In the text, you rather replace "must" with "should". Lines 46-47, the meaning is not very clear (40% of what?). In line 50, probably "the range of" is missing? In line 52, keep the unit to be referred once, as well as in line 61. In line 56 and 67, please provide references. Please, check if the term "cubic cube" is proper. In materials-methods chapter, provide the scientific names of the species used. Refer to the equipment used (for example to ground the materials, to mix them with glue etc.) including all the necessary details on the equipment (model, manufacturer, country). In line 88, clarify the "previously selected". Leave space between value and celsius unit. In line 92, you rather use "in terms of". You did not refer something on the conditioning of the panels after pressing completion. It would be useful to add also the number of specimens/repetitions for each property test. The standard ISO 10534-2:1998 has been updated in 2015. In 115-118 lines, do you believe that the number of 3 or 4 specimens would be adequate for the evaluation of mechanical strength of these composite materials (given the variety of factors affecting the final strength value obtained, for example pressure, temperature, moisture etc..)? please, explain, which standard has the number of specimens been based on? Add the method aaplied on the statistical analysis of the results. In the results, please add a comment on the fact that the length in fibres is necessary towards the achievement of the panel mechanical strength, while the incorporation of lignocellulosic materials in the form of powder in several matrices has achieved higher panel densities and lower thermal conductivity values compared to longer fibres (I propose to use the recent study of https://doi.org/10.1002/bbb.2291 to support such a statement). In figure 2 and 4, correct the "grups". In the tables, please provide as well the standard deviation values, except for the average values. In conclusions section, I think you could include a reference to your resukts concerning the optimum geometry of the straw particles size used in such panels.

Few grammatical and syntactical errors have been detected. Please, make a check in the whole text towards improvement. A rephrase is necessary to make the meanings clear in some cases highlighted in the above comments to the authors.

Author Response

Dear Reviewer

thank you for your help in editing this publication.

The topic of the research work and manuscript is really interesting and provides new and useful information. However there are some issues to be addressed towards its quality improvement before publication.

Thank you for your time and valuable suggestions.

In the abstract, you refer "thermal insulation of 0.033 - 0.046 W/(m-K)" and I wonder if the term thermal resistance would be more appropriate to be used.

It is corrected according to your suggestion.

In lines 35-38, as well as in the points that you refer to a patent or a study, please provide the most crucial and relevant to your topic findings (or for example in line 40, you refer reduced formaldehyde content without providing the reader how this was achieved).

Information has been supplemented as far as the patent description allows.

In the text, you rather replace "must" with "should".

It is corrected.

Lines 46-47, the meaning is not very clear (40% of what?).

It was a mistake which is corrected now. The patent description informs that straws with the length of 30-80 mm are required to achieve satisfactory properties of the boards. This information is now included.

In line 50, probably "the range of" is missing?

We agree, it is added.

In line 52, keep the unit to be referred once, as well as in line 61.

It is corrected.

In line 56 and 67, please provide references. Please, check if the term "cubic cube" is proper.

We added some reference, however, most of the information included was a result of a consult with industry representatives which came up with this issue, there is no reference to provide in this case. Term cubic cube was changed.

In materials-methods chapter, provide the scientific names of the species used. Refer to the equipment used (for example to ground the materials, to mix them with glue etc.) including all the necessary details on the equipment (model, manufacturer, country).

All necessary details are now included.

In line 88, clarify the "previously selected". Leave space between value and celsius unit.

It is clarified that it is selected based on the experience in work with these kinds of materials. The unit is corrected.

In line 92, you rather use "in terms of".

It is corrected.

You did not refer something on the conditioning of the panels after pressing completion.

The boards were of course conditioned, information about the conditions has been added.

It would be useful to add also the number of specimens/repetitions for each property test.

The number of repetitions is summarized in the last paragraph of the methodology.

The standard ISO 10534-2:1998 has been updated in 2015.

According to the information on the ISO website of 10534-2:1998, the standard was reviewed and confirmed in 2015 and therefore, this version remains current.

In 115-118 lines, do you believe that the number of 3 or 4 specimens would be adequate for the evaluation of mechanical strength of these composite materials (given the variety of factors affecting the final strength value obtained, for example pressure, temperature, moisture etc..)? please, explain, which standard has the number of specimens been based on?

The number of samples adopted does not follow the standard. The standards usually indicate eight specimens for the mechanical properties of wood-based materials. In the case of the tests presented in this publication, experience and the values of the test results obtained were the deciding factors. We have carried out numerous commercial tests for Westa-Eco and others. With a homogeneous material structure, which is how our boards are made, this number of samples is sufficient for the assumed differences in density. Of course, the greater the number, the more reliable the results. The density profile for this type of board is an indicative parameter, not an indicator of strength. Usually, such a measurement is carried out on three samples.

Add the method aaplied on the statistical analysis of the results.

It is included. The results of the study were analysed using ANOVA analysis of variance, while significance of differences between the groups was determined by Tukey's HSD test.

In the results, please add a comment on the fact that the length in fibres is necessary towards the achievement of the panel mechanical strength, while the incorporation of lignocellulosic materials in the form of powder in several matrices has achieved higher panel densities and lower thermal conductivity values compared to longer fibres (I propose to use the recent study of https://doi.org/10.1002/bbb.2291 to support such a statement). 

Following the suggestion of Reviewer 1, this information was included in the Introduction and Results and Discussion

In figure 2 and 4, correct the "grups".

It is corrected

In the tables, please provide as well the standard deviation values, except for the average values.

It is corrected.

In conclusions section, I think you could include a reference to your resukts concerning the optimum geometry of the straw particles size used in such panels.

Thank you for your valuable suggestion. Conclusions are corrected.

Round 2

Reviewer 2 Report

As I have checked the authors have implemented the proposed changes in the revised verion of manuscript towards the improvement of their work. Almost all the changes have been implemented and in my opinion, the manuscript is well-prepared and organized enough to be accepted for publication in this journal. 

English language use has been thorougly improved and only a brief check in the whole text for type errors is recommended.